# Effects of Low-Protein Diet Without Soybean Meal on Growth Performance, Nutrient Digestibility, Plasma Free Amino Acids, and Meat Quality of Finishing Pigs

**DOI:** 10.3390/ani15060828

**Published:** 2025-03-13

**Authors:** Bobo Deng, Litong Wang, Xiaomei Jiang, Tianyong Zhang, Mingfei Zhu, Guoshui Wang, Yizhen Wang, Yuanzhi Cheng

**Affiliations:** 1College of Animal Sciences, Zhejiang University, Hangzhou 310058, China; 18168960899@163.com (B.D.); yzwang321@zju.edu.cn (Y.W.); 2Zhejiang Tianpeng Group Co., Ltd., Quzhou 324014, China; 18367079999@163.com (L.W.); m18367070887@163.com (X.J.); ztyd2009@163.com (T.Z.); 13260642771@163.com (M.Z.)

**Keywords:** low-protein diet, soybean meal, growth performance, nitrogen utilization, meat quality, finishing pigs, gut health

## Abstract

This research investigated the impact of a low—protein diet without soybean meal on finishing pigs. The results showed that this diet didn’t adversely affect the pigs’ growth performance. It enhanced the digestibility of dry matter, crude protein, and calcium. Although there were some changes in plasma free amino acids, the meat quality of the pigs wasn’t negatively influenced. Moreover, this diet led to a reduction in ammonia—nitrogen and hydrogen sulfide levels in the pigs’ urine. In essence, feeding finishing pigs a low—protein diet without soybean meal can maintain their growth and meat quality while possibly cutting feed costs and lessening environmental impact.

## 1. Introduction

With the development of animal husbandry, the shortage of protein feed resources has become increasingly prominent, especially that of soybean meal as the main protein source, since its price fluctuation and supply stability have a great impact on the livestock industry [1,2]. Therefore, it is of great significance to study the application of a low-protein diet without soybean meal in pigs to reduce feed cost, improve feed utilization rate, and ensure the sustainable development of animal husbandry. The trial period was chosen to focus on the later stage of finishing pigs’ growth, which is an important period for assessing the impact of dietary changes on growth performance and meat quality. Soybean meal is one of the main protein raw materials in the pig diet, but with the rising price in recent years, soybean meal reduction alternatives have been paid more and more attention in China [3,4]. In pig production, low crude protein (CP) formulation with supplemental crystalline amino acid (CAA) has been widely implemented as a strategy to reduce the feed cost, and more recently to improve intestinal health mainly by decreasing the protein fermentation by intestinal microbiota [5,6,7]. This practice requires a higher energy-to-protein ratio in the diet to support rapid growth and fat deposition [8]. Therefore, a soybean meal-free diet with optimized protein levels can be particularly beneficial in this context. Reducing the dietary CP by 2% to 4% compared to the NRC (2012) [9] recommendations and supplementing with CAA has been demonstrated to increase nitrogen utilization, reduce feed costs and nitrogen excretion, and promote gut health without impairing the growth performance of pigs [10,11,12]. Robert et al. showed that reducing dietary CP content does not negatively impact the digestion of nutrients in the foregut of pigs [13]. Wu et al. reduced the dietary CP level of pigs by 1–3% and found that the growth performance of the pigs was not affected, and the NH_3_ content as well as nitrogen excretion in the fattening pig house were reduced regardless of the growth stage or fattening stage [14]. Li et al. showed that a 3% reduction in dietary CP level could increase the intramuscular fat content of the *longissimus dorsi* muscle of fattening pigs, reduce shear force, and help to improve meat quality [15]. However, significantly cutting down on dietary CP might negatively impact growth performance, as the synthesis of non-essential amino acids (AAs) and the availability of bioactive compounds from protein supplements could be restricted, even if the AA requirements are satisfied [16,17]. What’s more, using a low CP formula might lead to an increase in the net energy content of feeds, thereby causing excessive fat deposition [18]. On the other hand, adding more functional AAs to the low CP formula can further improve intestinal health and glucose metabolism, which in turn enhances nitrogen utilization and growth performance [19]. Although low-protein diets have been studied extensively in pigs, there are few studies based on low-protein diets without soybean meal. It is worth noting that in southern China, the selling price of 150 kg fattened pigs is significantly higher than that of 120 kg standard pigs, and breeding enterprises and family farms have a habit of raising large, fat pigs. According to the nutritional requirements of pigs, it is the most likely stage to use formula without soybean meal. Thus, a low-protein strategy can be realized.

There were few studies examining the application of a low-protein diet without soybean meal in the last stage of the fattening period; however, its effect on finishing pigs is still unknown. Hence, this work was conducted to evaluate the effects of a low-protein diet without soybean meal on growth performance, nutrient digestibility, plasma-free amino acids, and meat quality of finishing pigs. Together, these data could represent a useful basis for applications of a low-protein diet without soybean meal on finishing pigs to lay a good foundation for increasing efficiency and reducing cost.

## 2. Materials and Methods

A total of 81 barrows and 81 gilts ([Yorkshire × Landrace] × Duroc) of 150 days old with similar body weights (103.64 ± 3.65 kg) were selected and randomly allocated into three groups with 3 replicates in each group and 18 finishing pigs in each pen (4.5 m × 8.5 m). All pigs were housed in an environmentally controlled building with a half-slotted concrete floor facility during finishing periods. The experimental pigs were fed twice at 8:00 a.m. and 5:00 p.m. each day, and the epidemic prevention and disinfection were carried out following the management regulations of the pig farm. The temperature of the pig pens was controlled between 15 and 20 °C and the relative humidity was maintained between 50 and 65% throughout the experiment. Daily feed allocations to each pen were adjusted according to the body weight and the weight of the residual was recorded every day before new feeding was delivered. Fecal samples were collected three times daily at 8 h intervals. After collection, 10% H_2_SO_4_ was added to fix nitrogen, ammonium sulfate was used as marker, and the samples were stored at −20 °C for 24 h before processing.

The experiment was conducted at the Pig Experiment Research Farm, Zhejiang Tianpeng Group Co., Ltd. (Quzhou, China), from October to December 2023. Following the NRC guidelines from 1998 and 2012, three dietary formulations were established in this experiment (Table 1): (1) 14% crude protein, and 14% soybean meal for the control group diet (CON); (2) 13% crude protein, and 7% soybean meal for half soybean meal group diet (HSB); (3) 12% crude protein, and 0% soybean meal for soybean meal-free group diet (SBF). The three dietary groups were equal to standard ileal digestible amino acids and swine net energy. During the whole experiment period, each pen had feeders and nipple drinkers to provide ad libitum access. This experiment consisted of a 7-day adaptation and a 41-day fattening period for sample collection.

### 2.1. Sample Collection and Determination

The daily feed provided and remnants were collected and weighed to determine the ADFI. All 162 finishing pigs were weighed at the last day of the experiment to calculate final weight, average daily gain, and feed conversion rate. A week-long digestion and metabolism test was carried out at the conclusion of the formal period, with the first two days serving as the adaptation phase and the subsequent five days designated for sampling. Fecal samples were gathered three times daily, with an eight-hour interval between each collection. Subsequent to the collection, 10% sulfuric acid (H_2_SO_4_) was added to facilitate nitrogen fixation. The five-day fecal samples (200 g) were blended and stored in a refrigerator at 20 °C pending the assessment of the apparent digestibility of nutrients. Both feed and feces were subjected to drying and grinding using a pulverizer to prepare for the subsequent determination of dry DM, CP, EE, Ca, and P content. The detection of DM was performed in accordance with AOAC method 930.15; the measurement of EE was conducted following AOAC method 920.85; and the testing of CP was executed using the Kjeldahl method with an azotometer (Scino KT260, FOSS, Hillerod, Denmark).

At the conclusion of the experiment, six pigs with average body weights were chosen from each group for slaughter. Approximately 5 mL of blood was collected from the jugular vein of each pig into vacuum plasma tubes three hours after morning feeding [20]. One of the blood samples was then centrifuged at 3000× *g* for 10 min at 4 °C to obtain plasma, which was subsequently frozen at −20 °C for further analysis. Serum samples were analyzed for the levels of alanine aminotransferase (ALT), aspartate aminotransferase (AST), alkaline phosphatase (ALP), transglutaminase (GGT), total bilirubin (TBIL), total protein (TP), and so on, following the methods described in previous studies [21]. Another blood sample was used to measure the plasma-free amino acid content.

The weight of the carcass was recorded to calculate the carcass yield. Measurements of backfat thickness at the first rib, last rib, and last lumbar vertebra were taken to determine the average backfat depth. The loin eye area (LEA) was measured at the last lumbar vertebra [22]. Samples of the *longissimus dorsi* (LD) muscle from the 10th ribs were taken to measure the pH value, meat color, shear force, drip loss, cooking loss, moisture loss, and intramuscular fat (IMF) [23,24]. The ultimate pH was measured 24 h after slaughter using an Orion 9106 penetrating probe, which had been calibrated with two buffers (pH 7.00 and 4.01). Cooking loss was calculated 24 h after slaughter by comparing the weight of the cooked sample (B) to the weight of the raw sample (A) using the formula: cooking loss (g/100 g) = [(A − B)/A] × 100. Shear force, expressed as g/100 g of liquid expelled, was measured using the filter paper press method as described by the American Meat Science Association. Meat color was assessed at the same time and location as the pH measurement. The lightness (L*), redness (a*), and yellowness (b*) values were measured three times on the LD muscle for each pig using a Chroma meter (CR–400, Tokyo, Japan), and the average value was used as the result [25]. Shear force was measured using a texture analyzer (TA.XT Plus, Stable Micro Systems, Godalming, UK). The IMF was measured according to the national standard method (GB 5009.6–2016, China) [26]. Urine samples were collected from each pig at 3 h post-feeding. Ammonia-nitrogen levels were measured using the phenol–sodium hypochlorite colorimetric method, while hydrogen sulfide levels were determined using gas chromatography [27].

### 2.2. Statistical Analysis

The results are shown as the averages along with their standard errors of the mean (SEM). Statistical significance was assessed using one-way ANOVA followed by Tukey’s post–hoc tests for multiple comparisons. Before conducting ANOVA, data were tested for normality using the Shapiro–Wilk test and homogeneity of variance using Levene’s test. No pigs were eliminated or died during the experiment; hence, all data were included in the analysis. All analyses were performed using the SPSS Statistics software, version 20.0 (IBM Corp., Armonk, NY, USA). In this study, statistical significance was set at *p* ≤ 0.05, with highly significant results at *p* < 0.01, and trends were noted when 0.05 < *p* ≤ 0.1.

## 3. Results

### 3.1. Growth Performance

As shown in Table 2, the results showed no significant difference in FBW among all groups (*p* > 0.05). There was also no significant difference in ADFI and ADG among the CON, HSB, and SBF groups (*p* > 0.05). Although the corn content in the SBF group was significantly lower than in other groups, the pigs maintained similar growth performance, suggesting that the optimized protein and energy balance in the diet compensated for the reduced corn levels.

### 3.2. Nutrient Digestion and Serum Biochemical Parameters

As shown in Table 3, the results showed that compared with the CON group, the apparent digestibility of DM, CP, and Ca in the SBF group was significantly increased by 6.42%, 4.78%, and 9.82%, respectively (*p* < 0.05). However, EE and P did not differ significantly among the three groups (*p* > 0.05). As presented in Table 4, the results showed that there were no significant differences in different serum biochemical indexes among all groups (*p* > 0.05).

### 3.3. Plasma Free Amino Acid Composition

In this study, in order to explore the effects of low protein without soybean meal diet on plasma-free amino acids of 90–150 kg finishing pigs, the changes in 22 free amino acids in the plasma of CON, HSB, and SBF groups were compared (Table 5). Results demonstrated that compared with the CON group, the contents of glycine and glutamate in the HSB and SBF groups were significantly decreased by 42.90% and 37.06%, respectively (*p* < 0.05). Compared with the CON group, the contents of histidine and valine in the SBF group were significantly decreased by 16.72% and 16.21%, respectively (*p* < 0.05), but there was no significant difference compared with the HSB group (*p* > 0.05). Compared with the CON group, isoleucine content in the HSB group was significantly decreased by 50.39% (*p* < 0.05), but there was no significant difference compared with the SBF group (*p* > 0.05). The contents of other amino acids were not significantly different among the three groups (*p* > 0.05).

### 3.4. Meat Quality

As is shown in Table 6, backfat thickness in the SBF group was significantly increased by 3.53% compared with that in the CON group (*p* < 0.05), but there was no significant difference compared with the HSB group (*p* > 0.05). Compared with the CON group, lean meat percentage in the HSB and SBF groups tended to increase and decrease, respectively (*p* = 0.09). The indexes of slaughter rate and loin eye area showed no significant differences among the three groups (*p* > 0.05). There were no significant differences in pH value, meat color (a*, b*, L*), shear force, drip loss, cooking loss, milling loss, or intramuscular fat content indexes of *longissimus dorsi* muscle (*p* > 0.05).

### 3.5. The Content of H_2_S and Ammonia-Nitrogen in Urine

As is shown in Table 7, by adding no soybean meal to the diet of the SBF group, the urine ammonia-nitrogen content of 90–150 kg finishing pigs in the SBF group was significantly lower than that of the CON at the end of the experiment (*p* < 0.05), decreased by 28.80%. H_2_S content in the urine of the SBF group tended to decrease but there was no significant difference between the two groups (*p* = 0.06).

## 4. Discussion

### 4.1. Growth Performance, Nutrient Digestion, and Serum Biochemical Parameters

Growth performance is the most direct metric in livestock production. In this study, it was observed that feeding a low-protein diet, with crude protein reduced from 14% to 12%, had no negative impact on the growth performance of finishing pigs. Han et al. reported that in the growing stage, pigs fed a low-protein diet according to the NRC (2012) recommendations showed a decrease in average daily gain (ADG) [28]. Jiang et al. showed that supplementing glycine to a low-protein diet could improve meat quality without affecting animal growth [29]. Duan et al. found that when the amino acid requirements of growing–finishing pigs were met, reducing the dietary protein level by 2–4% did not affect the growth performance of the pigs [30]. Wang et al. demonstrated that pigs fed a low-protein diet might have longer villi in the small intestine during the finishing phase compared to those fed higher protein diets, which could lead to improved ADG [31,32]. Serum biochemical indicators can reflect changes in physiological functions and health status of the animal, as well as provide information on nutritional metabolism and organ function changes. The levels of serum albumin and total protein are also indicators of the animal’s health status [33]. In this study, we found that the low-protein diet without soybean meal had no negative effect on the blood biochemical parameters and thus the health status of finishing pigs. It should be noted that in this study, there was a significant substitution of maize with wheat in the diets. This change may have an impact on the results, as wheat and maize have different nutritional profiles and may affect the digestibility and utilization of nutrients. Future studies should consider this factor and investigate the effects of different grain sources on the performance of finishing pigs.

### 4.2. Free Amino Acid Profile in Plasma

The level of protein in the diet directly affects the metabolism of protein in the pig body, and reducing the level of protein in the diet can improve the metabolism of protein [33]. Amino acids are the basic components of protein and an important part of life activities. As an important metabolite of animal metabolism, plasma amino acids reflect the status of animal protein metabolism and energy metabolism [34]. Bergen et al. demonstrated that the content of free amino acids in plasma could reflect the metabolism of amino acids in animals. There was no significant difference in free amino acids in plasma, indicating that the ratio of various amino acids in the diet was balanced [35]. Ye et al. showed that after the reduction in dietary CP level, the plasma-free amino acids of finishing pigs were significantly reduced compared to that of the control group, except for histidine, valine, and isoleucine, which had similar content among groups [36]. In this study, the content of histidine and valine in plasma was decreased by the low-protein diet without soybean meal. It is indicated that the ratio of amino acids in the diet was basically balanced, and the metabolism and utilization of amino acids were not affected by the decrease of CP from 14 to 12%. The significant decrease in glycine and glutamate levels in the SBF group may reflect altered metabolic pathways in response to the low-protein diet. These changes could potentially affect energy metabolism and muscle development, although no adverse effects on growth performance were observed in this study. Especially synthetic AAs, such as L-lysine-HCl, DL-methionine, L-threonine, and tryptophan, were supplemented to increase nitrogen utilization and to reduce negative effects deriving from a reduction in CP. We hypothesized that when the above four amino acids are in sufficient levels in the diet, the reduction in crude protein would cause the finishing pigs to be more sensitive to the deficiency of histidine and valine.

### 4.3. Meat Quality and the Content of H_2_S and Ammonia-Nitrogen in Urine

There have been many reports on the influence of dietary protein levels on meat quality. The main indexes to evaluate meat quality are meat color, marbling, intramuscular fat, and pH value [37,38]. Li et al. found that lower levels of protein in the diet decreased shear force (improved tenderness), probably through an increase in total fat content and modification in fatty acid composition in the muscles of growing pigs [15]. Zhu et al. showed that a low-protein amino acid balanced diet could improve the tenderness of *longissimus dorsi* muscle, increase intramuscular fat content, and change the fatty acid composition of fattening pigs [39]. In our study, we found that backfat thickness in the SBF group was significantly increased by 3.53% compared with that in the CON group, but there were no significant differences in pH value, meat color (a*, b*, L*), shear force, drip loss, cooking loss, milling loss, and intramuscular fat content indexes of *longissimus dorsi* muscle when the proportion of soybean meal decreased from 14% to 0, indicating that low-protein diet without soybean meal did not have adverse effects on the meat quality of 90–150 kg finishing pigs. The levels of NH_3_-N and H_2_S in the urine not only affected the health of the livestock and humans but also reflected the nitrogen utilization capacity of animals [40,41]. In this study, NH_3_-N in the urine of finishing pigs fed with a low-protein diet without soybean meal was significantly decreased and the content of H_2_S in the excreta tended to be reduced.

## 5. Conclusions

In summary, under the conditions of this experiment, offering a low-protein diet without soybean meal to finishing pigs had no significant influence on the growth performance, but elevated the apparent digestibility of dry matter, crude protein, and calcium. In addition, there was no adverse effect on carcass traits, meat quality, and blood biochemical parameters on finishing pigs. Moreover, a low-protein diet without soybean meal reduced the emission of both ammoniacal nitrogen and sulfuretted hydrogen in urine. Therefore, a low-protein diet without soybean meal can be applied as a dietary strategy for finishing pigs, which can reduce the amount of soybean meal purchased annually for enterprises.

## Figures and Tables

**Table 1 animals-15-00828-t001:** Ingredients and nutritional contents of the experimental diets.

Items (%)	Treatment ^(1)^		
	CON	HSB	SBF
Ground corn	71.77	45.37	37.03
Wheat	-	25.00	30.00
Soybean meal	14.00	7.00	-
Wheat bran	12.00	10.00	10.00
Rice bran	-	5.00	5.00
Wheat germ meal	-	5.00	15.00
Soybean oil	0.50	0.50	0.50
Salt	0.40	0.40	0.40
Calcium carbonate	0.10	0.10	0.10
CaHPO_4_	0.50	0.50	0.50
Lysine	0.46	0.52	0.63
Methionine	0.03	0.04	0.05
Threonine	0.10	0.12	0.15
Tryptophan	-	0.01	0.12
Valine	-	-	0.02
Premix ^(2)^	0.14	0.44	0.50
Total	100.00	100.00	100.00
Nutrient Levels ^(3)^			
DM (%)	87.15	87.15	87.15
SNE (Kcal/kg)	2422.00	2418.00	2420.00
CP (%DM)	14.00	13.00	12.00
EE (%DM)	4.12	4.13	4.13
Ash (%DM)	3.50	3.51	3.53
Calcium (%DM)	0.58	0.58	0.58
Phosphorus (%DM)	0.24	0.24	0.24
Swine SID Lys (%)	0.90	0.90	0.90
Swine SID M + C (%)	0.45	0.45	0.45
Swine SID Met (%)	0.24	0.24	0.24
Swine SID Thr (%)	0.50	0.50	0.50
Swine SID Trp (%)	0.13	0.13	0.13
Swine SID Val (%)	0.53	0.53	0.53

Abbreviations: DM = dry matter; SNE = swine net energy; CP = crude protein; EE = ether extract; SID = standardized ileal digestibility; Lys = lysine; Met = methionine; C = cystine; Thr = threonine; Trp = tryptophan; and Val = valine. ^(1)^ Treatment: CON = 14% crude protein, and 14% soybean meal; HSB = 13% crude protein, and 7% soybean meal; SBF = 12% crude protein, and 0% soybean meal. ^(2)^ Premix was formulated to provide (per kilogram of the dietary DM) 31.5 mg of Zn as ZnSO_4_·7H_2_O; 40.1 mg of Mn as MnSO_4_·H_2_O; 90.0 mg of Fe as FeSO_4_·7H_2_O; 12.0 mg of Cu as CuSO_4_·5H_2_O; 0.5 mg of I as KI; 0.3 mg of Co as CoCl_2_·6H_2_O; 0.25 mg of Se as Na_2_SeO_3_; 6800 IU of vitamin A; 4500 IU of vitamin D_3_; 200 IU of vitamin E; 9 mg of vitamin K_3_, 5.88 mg of vitamin B_1_, 4.5 mg of vitamin B_6_; and 40 μg of vitamin B_12_. ^(3)^ Swine net energy and SID amino acid were calculated value, while the others were measured values.

**Table 2 animals-15-00828-t002:** Influence of low-protein diet without soybean meal on growth performance of finishing pigs.

Items		Treatment		SEM	*p*-Value
	CON (*n* = 3)	HSB (*n* = 3)	SBF (*n* = 3)		
IBW (kg)	103.28	104.30	103.35	5.03	0.35
FBW (kg)	151.26	153.43	151.48	8.46	0.42
ADFI (g/d)	4209.57	4230.20	4214.09	35.15	0.61
ADG (g/d)	1170.28	1198.25	1173.89	13.10	0.14
FCR	3.60	3.53	3.59	0.02	0.08

Abbreviations: IBW = initial body weight; FBW = final body weight; ADFI = average daily feed intake; ADG = average daily gain; FCR = feed conversion rate.

**Table 3 animals-15-00828-t003:** Influence of low-protein diet without soybean meal on nutrient digestion of finishing pigs.

Items		Treatment		SEM	*p*-Value
	CON (*n =* 6)	HSB (*n =* 6)	SBF (*n =* 6)		
DM (%)	80.10 ^b^	82.28 ^ab^	86.25 ^a^	2.13	<0.05
CP (%)	79.50 ^b^	81.10 ^ab^	83.45 ^a^	1.05	<0.05
EE (%)	75.65	74.60	75.31	3.40	0.38
Ca (%)	55.20 ^b^	56.13 ^b^	60.62 ^a^	2.25	<0.05
P (%)	47.24	51.50	50.68	3.62	0.72

Abbreviations: DM = dry matter; CP = crude protein; EE = ether extract; Ca = calcium; and P = phosphorus. ^a,b^ Means within a row with different superscripts differ (*p* < 0.05). The same as in the following table.

**Table 4 animals-15-00828-t004:** Influence of low-protein diet without soybean meal on serum biochemical parameters of finishing pigs.

Items		Treatment		SEM	*p*-Value
	CON (*n =* 6)	HSB (*n =* 6)	SBF (*n =* 6)		
ALT (U/L)	50.83	52.03	49.96	6.02	0.25
AST (U/L)	35.13	35.30	32.30	3.79	0.42
ALP (U/L)	70.67	71.60	72.33	5.50	0.38
GGT (U/L)	40.00	40.33	42.67	3.26	0.16
TBIL (μmoI/L)	7.70	5.23	6.93	0.65	0.22
TP (g/L)	74.73	68.96	75.96	5.32	0.57
ALB (g/L)	41.37	41.26	43.16	3.20	0.32
GLO (g/L)	33.36	27.70	32.80	5.58	0.46
A/G	1.26	1.50	1.33	0.35	0.78
LDH (U/L)	601.00	610.67	607.67	25.65	0.35
CK (U/L)	1426.33	1590.00	1500.67	100.20	0.92
BUN (mmoI/L)	4.91	4.54	5.76	0.80	0.20
CREA(μmoI/L)	149.20	140.50	146.30	10.25	0.33
GLU (mmoI/L)	5.93	6.03	6.23	0.50	0.53
CHOL (mmoI/L)	2.31	1.99	2.35	0.04	0.12
TG (mmoI/L)	0.43	0.35	0.45	0.08	0.16
PAMY (U/L)	2236.33	2424.00	2380.33	158.35	0.87
LIP (U/L)	17.76	14.70	16.43	3.20	0.28

Abbreviations: ALT = glutamate pyruvate transaminase; AST = glutamic oxalacetic transaminase; ALP = alkaline phosphatase; GGT = transglutaminase; TBIL = total bilirubin; TP = total protein; ALB = albumin; GLO = globulin; A/G = albumin to globulin; LDH = lactic dehydrogenase; CK = creatine kinase; BUN = urea nitrogen; CREA = creatinine; GLU = glucose; CHOL = total cholesterol; TG = triglyceride; PAMY = amylopsin; and LIP = lipase.

**Table 5 animals-15-00828-t005:** Influence of low-protein diet without soybean meal on plasma-free amino acid composition in finishing pigs.

Items		Treatment		SEM	*p*-Value
	CON (*n =* 6)	HSB (*n =* 6)	SBF (*n =* 6)		
Glycine (μg/mL)	219.10 ^a^	125.10 ^b^	137.91 ^b^	10.03	<0.05
Serine (μg/mL)	253.04	248.56	246.21	6.46	0.12
Methionine (μg/mL)	72.33	63.64	66.59	5.15	0.51
Proline (μg/mL)	28.36	25.72	24.37	3.20	0.24
Leucine (μg/mL)	33.55	30.78	31.64	2.12	0.78
Creatine (μg/mL)	43.71	33.76	42.53	6.28	0.37
Glutamate (μg/mL)	229.20 ^a^	165.13 ^b^	173.69 ^b^	10.20	<0.05
Phenylalanine (μg/mL)	18.67	12.89	16.64	3.50	0.13
Lysine (μg/mL)	56.96	52.71	54.34	3.78	0.62
Argine (μg/mL)	25.39	21.86	20.89	4.10	0.39
Tryptophan (μg/mL)	15.18	12.85	15.76	2.30	0.17
Tyrosine (μg/mL)	13.01	11.49	12.02	1.80	0.59
Histidine (μg/mL)	47.67 ^a^	43.06 ^ab^	39.70 ^b^	0.90	<0.05
Valine (μg/mL)	142.68 ^a^	138.52 ^ab^	119.55 ^b^	8.15	<0.05
Ornithine (μg/mL)	57.42	50.09	52.12	6.05	0.17
Alanine (μg/mL)	40.59	33.83	39.92	5.26	0.82
Taurine (μg/mL)	31.83	31.99	35.33	3.80	0.28
Isoleucine (μg/mL)	23.10 ^a^	11.46 ^b^	17.23 ^ab^	4.30	<0.05
Aspartic (μg/mL)	10.52	8.61	9.03	1.28	0.47
Threonine (μg/mL)	42.64	39.46	33.93	5.90	0.62
Glutamine (μg/mL)	58.21	54.72	56.06	3.02	0.15
Asparagine(μg/mL)	13.89	14.62	16.27	2.40	0.17

^a,b^ Means within a row with different superscripts differ (*p* < 0.05). The same as in the following table.

**Table 6 animals-15-00828-t006:** Influence of low-protein diet without soybean meal on carcass traits and meat quality of finishing pigs.

Items		Treatment		SEM	*p*-Value
	CON (*n =* 6)	HSB (*n =* 6)	SBF (*n =* 6)		
Carcass traits					
YP (%)	72.83	70.25	71.95	1.32	0.56
CL (cm)	91.05	90.50	92.89	2.58	0.42
BFT (cm)	2.83 ^b^	2.87 ^ab^	2.93 ^a^	0.05	<0.05
LEA (cm^2^)	32.58	33.06	32.73	2.80	0.17
LMR (%)	60.20	61.52	57.90	2.35	0.09
Meat quality					
pH_24h_	5.72	5.68	5.70	0.15	0.35
a*	16.30	16.25	16.81	1.80	0.57
b*	4.23	4.50	4.46	0.45	0.81
L*	56.10	57.20	56.52	3.55	0.70
SF (N)	37.00	36.50	37.87	4.90	0.32
DL (%)	4.65	4.70	4.62	0.89	0.61
CL (%)	15.50	14.92	15.10	0.80	0.20
ML (%)	37.10	37.45	38.06	10.25	0.38
IMF (%)	1.80	1.84	1.95	0.50	0.53

Abbreviations: YP = yield percentage; CL = carcass length; BFT = back fat thickness; LEA = loin eye area; LMR = lean meat rate; SF = shear force; DL = drip loss; CL = cooking loss; ML = milling loss; and IMF = intramuscular fat. ^a,b^ Means within a row with different superscripts differ (*p* < 0.05). The same as in the following table.

**Table 7 animals-15-00828-t007:** Influence of low-protein diet without soybean meal on the content of H_2_S and ammonia-nitrogen in urine of finishing pigs.

Items		Treatment		SEM	*p*-Value
	CON (*n =* 6)	HSB (*n =* 6)	SBF (*n =* 6)		
NH_3_-N (mg/dL)	1.25 ^a^	1.14 ^ab^	0.89 ^b^	0.13	<0.05
H_2_S (nmol/mL)	5.25	5.16	4.12	0.75	0.06

^a,b^ Means within a row with different superscripts differ (*p* < 0.05). The same as in the following table.

## Data Availability

The original contributions presented in this study are included in the article. Further inquiries can be directed to the corresponding author(s).

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
