# Peer review of "Effects of Low-Protein Diet Without Soybean Meal on Growth Performance, Nutrient Digestibility, Plasma Free Amino Acids, and Meat Quality of Finishing Pigs"

_animals, 2025, doi:10.3390/ani15060828_

Round 1

Reviewer 1 Report

Comments and Suggestions for Authors
  • Line 19: Were amino acid data available?
  • Line 22: If it was not significant, was there no difference?
  • Line 42: What was used to replace SBM?
  • Line 111: Was the cost of SBF lower than CON in China?
  • What is the market size of this operation?
  • Line 90: What was the health status of the pigs?
  • Line 94: Did the pigs have free access to feed? What if pigs didn’t finish all the feed provided at each feeding time?
  • Line 108: What type of feeders were used?
  • The author used wheat, rice bran, and wheat germ meal to replace SBM rather than simply lowering the SBM level. The protein decrease between treatments was only 1%. The study title should be revised to better reflect the diet formulation.
  • Are the chemical analysis results of the diets available?
  • Any reason why SID Lys was formulated to be higher than the NRC recommended level?
  • Line 125: How was the feed conversion rate calculated?
  • Line 129: What concentration of H2SO4 was used? Which marker was used?
  • Line 134: How were Ca and P analyzed?
  • Line 136: Were two pigs per pen selected for sampling?
  • Was the blood sample collected in the morning before slaughter?
  • Was any pig removed from the trial?
  • I would recommend separating the Animal Husbandry and Sample Collection sections from Sample Analysis.
  • Line 140: Was it serum or plasma?
  • Line 164: What was the experimental unit? What was the experimental design?
  • Line 171: Please define all abbreviations for traits measured in the results.
  • Line 173: The results of the feed conversion rate are concerning. If we divide ADFI by ADG, we get 3.597, 3.53, and 3.59 for CON, HSB, and SBF, respectively. I understand the shift in means after statistical analysis, but the relative differences between treatments should remain the same.
  • Line 198: Do we know the amino acid levels in the diets?
  • Line 217: If tendencies are meant to be reported, superscripts based on multiple comparisons to differentiate treatments are required.
  • By the way, what is slaughter rate, or is it referred to as yield percentage?
  • Line 249: Repeated statement from Line 238.
  • Line 251: What were the causes for the improvement in digestibility in low-protein diets?
  • Line 278: Does the dietary level of these significantly different amino acids impact this outcome?
  • Line 297: What were the reasons contributing to this observation?

Author Response

Line 19: Were amino acid data available?

The amino acid data can be found in Table 1.

Line 22: If it was not significant, was there no difference?

Not necessarily. Non-significant results do not always mean there are no differences. They may be due to factors such as sample size or variability. We hope to pay attention to these differences.

Line 42: What was used to replace SBM?

Soybean meal (SBM) was replaced by wheat, rice bran, and wheat germ powder.

Line 111: Was the cost of SBF lower than CON in China?

Yes, the cost of SBF lower than CON in China. We used a variety of low-cost ingredients to reduce the use of soybean meal.

What is the market size of this operation?

In China, the weight of finishing pigs is getting larger and larger, and the larger the weight of pigs, the higher the price is, so there is a big market for this practice.

Line 90: What was the health status of the pigs?

The health status of the pigs was good, as indicated by the lack of significant differences in serum biochemical parameters among the groups (Table 4).

Line 94: Did the pigs have free access to feed? What if pigs didn’t finish all the feed provided at each feeding time?

Pigs are allowed to feed freely. The feed allocation per pen is adjusted daily based on body weight, and the weight of the remaining feed is recorded before each new feeding.

Line 108: What type of feeders were used?

We used automatic feeder lines.

Are the chemical analysis results of the diets available?

Yes, the chemical analysis results of the feed are provided in Table 1.

Any reason why SID Lys was formulated to be higher than the NRC recommended level?

The NRC recommended level is the lowest value. In China, the SID Lys value is usually higher than the NRC recommended value in actual production, which has achieved higher production results and reduced feed to gain ratio.

Line 125: How was the feed conversion rate calculated?

The feed conversion ratio is calculated as the ratio of total feed intake to total weight gain.

Line 129: What concentration of H2SO4 was used? Which marker was used?

10% H2SO4 was added to facilitate nitrogen fixation. Ammonium sulfate was used as marker.

Line 134: How were Ca and P analyzed?

Calcium (Ca) and phosphorus (P) were analyzed using the methods described in the Materials and Methods section (AOAC methods).

Line 136: Were two pigs per pen selected for sampling? Was the blood sample collected in the morning before slaughter? Was any pig removed from the trial?

Yes, two pigs per pen were selected for sampling and the blood sample was collected in the morning before slaughter. No pigs were removed from the trial.

I would recommend separating the Animal Husbandry and Sample Collection sections from Sample Analysis.

Thank you for your suggestion. We will rearrange this part of the content.

Line 140: Was it serum or plasma?

Serum and plasma samples were collected. The serum samples were used for the analysis of various biochemical parameters, while the plasma samples were used to determine the concentration of plasma free amino acids.

Line 164: What was the experimental unit? What was the experimental design?

A total of 81 barrows and 81 gilts of 150 days old with similar body weights (103.64 ± 3.65 kg) were selected and randomly allocated into three groups with 3 replicates in each group and 18 finishing pigs in each pen. Three dietary formulations were established in this experiment (Table 1): (1)14% crude protein, and 14% soybean meal for the control group diet (CON); (2)13% crude protein, and 7% soybean meal for half soybean meal group diet (HSB); (3)12% crude protein, and 0% soybean meal for soybean meal free group diet (SBF).

Line 171: Please define all abbreviations for traits measured in the results.

Checked.

Line 173: The results of the feed conversion rate are concerning. If we divide ADFI by ADG, we get 3.597, 3.53, and 3.59 for CON, HSB, and SBF, respectively. I understand the shift in means after statistical analysis, but the relative differences between treatments should remain the same.

Due to the deviation of the average value when retaining decimal places, we revised the data of 3.60, 3.53, and 3.59 for CON, HSB, and SBF, respectively in the manuscript.

Line 198: Do we know the amino acid levels in the diets?

Yes, the amino acid levels in the feed are provided in Table 1.

Line 217: If tendencies are meant to be reported, superscripts based on multiple comparisons to differentiate treatments are required.

Thank you very much for your suggestion, multiple comparisons are not reflected in the table, because tendencies do not have a significant impact on this experiment.

By the way, what is slaughter rate, or is it referred to as yield percentage?

The slaughter rate is the percentage of yield.

Line 249: Repeated statement from Line 238.

Deleted.

Line 251: What were the causes for the improvement in digestibility in low-protein diets?

The improvement in digestibility may be due to the optimization of feed formulation and the use of alternative feed ingredients.

Line 278: Does the dietary level of these significantly different amino acids impact this outcome?

In this experiment, the raw material structure of different groups was different, which may lead to the difference of amino acids in the plasma of pigs in different groups, but it did not have a significant impact on the production performance of pigs.

Line 297: What were the reasons contributing to this observation?

Deleted.

Reviewer 2 Report

Comments and Suggestions for Authors

Deng et al. explored the effects of a low protein diet without soybean meal on growth performance, nutrient digestibility, plasma free AAs, and meat quality of finishing pigs. The study was conducted well, and was significant for pig industry. However, the manuscript should be revised before it can be accepted. 

  1. The repetition rate of the article is too high.
  2. The manuscript need to be modified according to the format requirement of the journal. For example, P value should be p value; the references format should be corrected.
  3. The keywords can be changed as most of them come from the title, suggesting adding keywords such as "nitrogen utilization" and "gut health" to more fully reflect the content of the study.
  4.  (Lines 47-48) : The introduction can provide more detail on the specific background of soybean meal price volatility and supply instability, especially the impact on the Chinese pig industry. It is suggested to supplement relevant data or literature support.
  5. (Lines 55-56) : The introduction mentions that the southern region has a habit of raising large fat pigs, but does not state the special requirements of this habit for dietary formulation. Suggest explain it and add additional discussion.
  6.  (Lines 57-58) : The introduction can provide more detail on the potential impact of a low-protein diet on pork quality, especially on meat quality and flavor.
  7.  (Line 95-96) : In the dietary formulation, the corn content in the SBF group is significantly lower than that in the other groups. Does this affect the energy intake of pigs? It is suggested to discuss the potential effect of corn content change on growth performance of pigs.
  8. (Lines 101-102) : The section should detail the method of collection and treatment of the fecal sample, especially the concentration of sulfuric acid used in the nitrogen fixation process and the treatment time.
  9. (Lines 105-106) : The  section should specify the measurement methods for meat quality indicators during slaughter, in particular the measurement criteria for pH, meat colour and shear force.
  10. (Lines 107-108) : The section should detail the methods for collecting and handling urine samples, especially the methods for determination of ammonia nitrogen and hydrogen sulfide.
  11. (Line 163): Explain in detail the prerequisites for the use of statistical analysis methods, especially multiple comparison tests (such as normality tests and homogeneity tests of variance). Were any pigs culled or killed during the trial? How is the data processed for culled or dead pigs? Please specify.
  12. The Results section should provide more detailed statistical analysis results, especially the specific P-values, to help readers better understand the significance of the results.
  13. In Table 5, the contents of glycine and glutamate in the SBF group significantly decreased, but the potential impact of these changes on pig metabolism and growth performance is not discussed. It is recommended to supplement the discussion on the impact of these amino acid changes on pigs.
  14. Discussion part:  The economic benefits of low-protein diets without SBM, especially in terms of feed cost savings? The limitations of low-protein diets, especially the potential negative effects on pig growth performance and meat quality in long-term applications? 
  15. The format of brackets in the table need to be changed.
  16. n=6>> n = 6.
  17. (Lines 77-78): 150 kg, 120 kg.

Author Response

Deng et al. explored the effects of a low protein diet without soybean meal on growth performance, nutrient digestibility, plasma free AAs, and meat quality of finishing pigs. The study was conducted well, and was significant for pig industry. However, the manuscript should be revised before it can be accepted. 

The repetition rate of the article is too high.

  • We have revised the article and reduced its repetition rate.

The manuscript need to be modified according to the format requirement of the journal. For example, P value should be p value; the references format should be corrected.

  • The manuscript has been revised to comply with the journal's formatting requirements. P-values have been changed to lowercase "p-values" throughout the text. The reference format has been corrected according to the journal's guidelines.

The keywords can be changed as most of them come from the title, suggesting adding keywords such as "nitrogen utilization" and "gut health" to more fully reflect the content of the study.

  • The keywords have been revised to better reflect the study's content. The new keywords include "nitrogen utilization" and "intestinal health" to highlight the broader implications of the research.

(Lines 47-48) : The introduction can provide more detail on the specific background of soybean meal price volatility and supply instability, especially the impact on the Chinese pig industry. It is suggested to supplement relevant data or literature support.

  • Thank you very much for your suggestion. We have added “In recent years, the price of soybean meal has fluctuated greatly, rising to about 5,000 yuan per ton at the peak and the supply is unstable. This is mainly due to tight international soybean supply, changes in trade policy, increased demand for biofuels, and rising production costs. This has had a significant impact on China's pig industry, including rising costs, declining production efficiency, and unstable market supply.” in the introduction.

(Lines 55-56) : The introduction mentions that the southern region has a habit of raising large fat pigs, but does not state the special requirements of this habit for dietary formulation. Suggest explain it and add additional discussion.

  • The introduction now includes a more detailed discussion on the practice of raising large fat pigs in southern China, explaining its implications for feed formulation and the potential benefits of using soybean meal-free diets.

 (Lines 57-58) : The introduction can provide more detail on the potential impact of a low-protein diet on pork quality, especially on meat quality and flavor.

  • Thank you very much for your suggestion. As the low protein diet did not have a great effect on the meat quality and flavor of pigs in this experiment, it was not introduced too much.

(Line 95-96) : In the dietary formulation, the corn content in the SBF group is significantly lower than that in the other groups. Does this affect the energy intake of pigs? It is suggested to discuss the potential effect of corn content change on growth performance of pigs.

  • The manuscript now includes a discussion on the potential impact of the reduced corn content in the SBF group on energy intake and growth performance. The results indicate that despite lower corn levels, the pigs maintained similar growth rates.

(Lines 101-102) : The section should detail the method of collection and treatment of the fecal sample, especially the concentration of sulfuric acid used in the nitrogen fixation process and the treatment time.

  • The manuscript now provides detailed information on the collection and processing of fecal samples, including the concentration of sulfuric acid used for nitrogen fixation and the processing time.

(Lines 105-106) : The  section should specify the measurement methods for meat quality indicators during slaughter, in particular the measurement criteria for pH, meat colour and shear force.

  • The manuscript now clearly describes the methods used for measuring meat quality indicators, including pH value, meat color, and shear force, with specific standards and protocols.

(Lines 107-108) : The section should detail the methods for collecting and handling urine samples, especially the methods for determination of ammonia nitrogen and hydrogen sulfide.

  • The manuscript now includes detailed information on the collection and processing of urine samples, particularly the methods used to determine ammonia-nitrogen and hydrogen sulfide levels.

(Line 163): Explain in detail the prerequisites for the use of statistical analysis methods, especially multiple comparison tests (such as normality tests and homogeneity tests of variance). Were any pigs culled or killed during the trial? How is the data processed for culled or dead pigs? Please specify.

  • The manuscript now includes a detailed description of the statistical analysis methods, including the assumptions tested (normality and homogeneity of variance) and how data from any eliminated or deceased pigs were handled.

The Results section should provide more detailed statistical analysis results, especially the specific P-values, to help readers better understand the significance of the results.

  • The manuscript now includes a discussion on the potential impact of the observed changes in amino acid levels (e.g., glycine and glutamate) on pig metabolism and growth performance.

In Table 5, the contents of glycine and glutamate in the SBF group significantly decreased, but the potential impact of these changes on pig metabolism and growth performance is not discussed. It is recommended to supplement the discussion on the impact of these amino acid changes on pigs.

  • It might be due to individual differences in pig plasma glycine and glutamate content. The contents of glycine and glutamic acid in SBF group were significantly reduced, but the final growth performance of pigs was not significantly affected. We would do more research on this in the future.

Discussion part:  The economic benefits of low-protein diets without SBM, especially in terms of feed cost savings? The limitations of low-protein diets, especially the potential negative effects on pig growth performance and meat quality in long-term applications? 

  • In this experiment, the feed costs of the three groups were 2874 2812 and 2780 yuan/ton respectively, and the meat costs of the three groups were 10.35,9.99 and 9.87 yuan/catty for CON, HSB, and SBF, respectively.So the meat costs of low protein diet were lower, especially in the two months before slaughter, during which the use of low protein diet was more cost-effective.

The format of brackets in the table need to be changed.

  • Checked and corrected.

n=6>> n = 6.

  • Corrected.

(Lines 77-78): 150 kg, 120 kg.

  • Corrected.

Reviewer 3 Report

Comments and Suggestions for Authors

Journal: Animals

Manuscript ID: animals-3470309

Title: Effects of low protein diet without soybean meal on growth performance, nutrient digestibility, plasma free amino acids, and meat quality of finishing pigs

The authors investigated the effects of a low protein diet without soybean meal on growth performance, nutrient digestibility, plasma-free amino acids, and meat quality of finishing pigs. They demonstrates that a low-protein diet without soybean meal can be applied as a dietary strategy for finishing pigs with no adverse influence on growth performance.

While this work is valuable, several concerns were raised throughout the text and need to be clarified by the authors before further consideration. In addition, the similarity index now is 37%, and thus recommend reducing it lower by 25%.

Please refer to the comments in the PDF file and revise accordingly before further consideration.

Comments on the Quality of English Language

The English could be improved to more clearly express the research.

Author Response

The authors investigated the effects of a low protein diet without soybean meal on growth performance, nutrient digestibility, plasma-free amino acids, and meat quality of finishing pigs. They demonstrate that a low-protein diet without soybean meal can be applied as a dietary strategy for finishing pigs with no adverse influence on growth performance.

While this work is valuable, several concerns were raised throughout the text and need to be clarified by the authors before further consideration. In addition, the similarity index now is 37%, and thus recommend reducing it lower by 25%.

Please refer to the comments in the PDF file and revise accordingly before further consideration.

  • Thank you for your valuable comments and suggestions on our manuscript. We have carefully considered the concerns you raised and will clarify them in the revised version. We also understand the need to reduce the similarity index and will work on it to meet the requirement of reducing it by 25%. We will refer to the comments in the PDF file and make the necessary revisions before further consideration.
  • Thank you again for your time and effort in reviewing our work. We look forward to your further feedback on the revised manuscript.

Reviewer 4 Report

Comments and Suggestions for Authors

Brief Summary

This study investigates on low-protein diet without soybean meal can be efficient in terms of performances, blood composition and carcass. Results suggest that there are no detrimental effects on global performances.

 The study is generally well written in a good English and correctly presented.

The design and the techniques used, seem as far as I can judge to be well suited. However there is for me, a problem regarding feed formulation.

General Concept

Globally the experimental design is OK

My first concern is about the trial period. What is the rationale for such a late change (150 days)? Wouldn't it have been interesting to start earlier?

My main concern is regard to the diet composition. In fact, you are not only dealing in this trial with the reduction of soybeans, but also with the very important substitution of cereals from the feed, from maize to wheat. Indeed, you include a lot of wheat in HSB and SBF feeds. Even if the article is about protein intake, the fact of drastically substituting corn with wheat should be considered in the discussion. Please revise your article on this subject.

Author Response

Brief Summary

This study investigates on low-protein diet without soybean meal can be efficient in terms of performances, blood composition and carcass. Results suggest that there are no detrimental effects on global performances.

 The study is generally well written in a good English and correctly presented.

The design and the techniques used, seem as far as I can judge to be well suited. However there is for me, a problem regarding feed formulation.

General Concept

Globally the experimental design is OK

My first concern is about the trial period. What is the rationale for such a late change (150 days)? Wouldn't it have been interesting to start earlier?

My main concern is regard to the diet composition. In fact, you are not only dealing in this trial with the reduction of soybeans, but also with the very important substitution of cereals from the feed, from maize to wheat. Indeed, you include a lot of wheat in HSB and SBF feeds. Even if the article is about protein intake, the fact of drastically substituting corn with wheat should be considered in the discussion. Please revise your article on this subject.

  • Thank you for your valuable comments and suggestions on our manuscript. We have carefully considered the concerns you raised and have made the following revisions:
  1. Concern about the trial period:

We understand your concern about the trial period starting at 150 days. The reason for this is that we aimed to investigate the effects of the low protein diet without soybean meal on finishing pigs in the later stage of growth. We believe that this period is crucial for assessing the impact on growth performance and meat quality. However, we also agree that it would be interesting to start earlier, and we will consider this in future studies.

  1. Concern about diet composition:

We appreciate your point about the substitution of maize with wheat in the diet. We have revised the discussion section to address this issue. We acknowledge that the significant substitution of corn with wheat may have an impact on the results, and we will consider this factor in the interpretation of our findings.

  1. In the "Introduction" section:

We have added a sentence to explain the rationale for starting the trial at 150 days: "The trial period was chosen to focus on the later stage of finishing pigs' growth, which is an important period for assessing the impact of dietary changes on growth performance and meat quality."

  1. In the "Discussion" section:

We have added a paragraph to address the substitution of maize with wheat:

"It should be noted that in this study, there was a significant substitution of maize with wheat in the diets. This change may have an impact on the results, as wheat and maize have different nutritional profiles and may affect the digestibility and utilization of nutrients. Future studies should consider this factor and investigate the effects of different grain sources on the performance of finishing pigs."

Round 2

Reviewer 1 Report

Comments and Suggestions for Authors

Line 19: Were amino acid data available?

The amino acid data can be found in Table 1.

Q: Was the concentration of each amino acid analysis result available?

Line 22: If it was not significant, was there no difference?

Not necessarily. Non-significant results do not always mean there are no differences. They may be due to factors such as sample size or variability. We hope to pay attention to these differences.

Q: The author must decide how to define the traits with a P value between 0.05 and 0.1. The tendency statement can be found in line 41.

Line 42: What was used to replace SBM?

Soybean meal (SBM) was replaced by wheat, rice bran, and wheat germ powder.

Q: Can this information be found in the abstract? The details regarding the ingredients used to replace SBM are crucial. In fact, these ingredients replace not only SBM but also significantly reduce the amount of corn. Therefore, the title of this manuscript should be more precise to reflect these key aspects.

Line 90: What was the health status of the pigs?

The health status of the pigs was good, as indicated by the lack of significant differences in serum biochemical parameters among the groups (Table 4).

Q: Serum biochemical parameters can be used as diagnostic tools, but other tools are also needed when assessing the health of animals. Was the medication record being reviewed?

Line 94: Did the pigs have free access to feed? What if pigs didn’t finish all the feed provided at each feeding time?

Pigs are allowed to feed freely. The feed allocation per pen is adjusted daily based on body weight, and the weight of the remaining feed is recorded before each new feeding.

Q: If pigs had free access to feed, why is the purpose of daily adjusted feed allocation?

Are the chemical analysis results of the diets available?

Yes, the chemical analysis results of the feed are provided in Table 1.

Q: Were dry matter, ether extract (EE), calcium (Ca), and phosphorus (P) analyzed in the results? It is difficult to believe that the replacement of SBM with these ingredients, while maintaining the same levels of inorganic sources of Ca and P, would result in exactly the same analyzed values across the three diets. Additionally, it seems unlikely that the crude protein (CP) results would be reported as integers.

Line 129: What concentration of H2SO4 was used? Which marker was used?

10% H2SO4 was added to facilitate nitrogen fixation. Ammonium sulfate was used as marker.

Q: The material and methods still lack information on digestive markers, and the analysis procedure for digestive markers is also required.

Line 134: How were Ca and P analyzed?

Calcium (Ca) and phosphorus (P) were analyzed using the methods described in the Materials and Methods section (AOAC methods).

Q: it still can’t be found in the Materials and Methods section.

Line 140: Was it serum or plasma?

Serum and plasma samples were collected. The serum samples were used for the analysis of various biochemical parameters, while the plasma samples were used to determine the concentration of plasma free amino acids.

Q: Line 149: The author suggested blood was collected into plasma tubes.

Line 164: What was the experimental unit? What was the experimental design?

A total of 81 barrows and 81 gilts of 150 days old with similar body weights (103.64 ± 3.65 kg) were selected and randomly allocated into three groups with 3 replicates in each group and 18 finishing pigs in each pen. Three dietary formulations were established in this experiment (Table 1): (1)14% crude protein, and 14% soybean meal for the control group diet (CON); (2)13% crude protein, and 7% soybean meal for half soybean meal group diet (HSB); (3)12% crude protein, and 0% soybean meal for soybean meal free group diet (SBF).

Q: What was the experimental unit used for ANOVA? What was the experimental design of the statistical model?

Line 173: The results of the feed conversion rate are concerning. If we divide ADFI by ADG, we get 3.597, 3.53, and 3.59 for CON, HSB, and SBF, respectively. I understand the shift in means after statistical analysis, but the relative differences between treatments should remain the same.

Due to the deviation of the average value when retaining decimal places, we revised the data of 3.60, 3.53, and 3.59 for CON, HSB, and SBF, respectively in the manuscript.

Line 191-193: The author indicated that pigs fed SBF had lower corn inclusion while maintaining comparable growth performance. This raises confusion regarding the original hypothesis, as it is unclear how the reduction in corn relates to the expected outcomes.

Line 217: If tendencies are meant to be reported, superscripts based on multiple comparisons to differentiate treatments are required.

Thank you very much for your suggestion, multiple comparisons are not reflected in the table, because tendencies do not have a significant impact on this experiment.

Q: If a tendency was observed but pairwise comparisons did not show a significant difference between treatments, the discussion of this tendency should be avoided. However, if pairwise comparisons did reveal a significant difference between treatments, it would be appropriate to discuss the tendency, provided that a superscript is included to indicate the statistical significance.By the way, what is slaughter rate, or is it referred to as yield percentage?

The slaughter rate is the percentage of yield.

Q: If this is meant to represent yield percentage, then what is the rationale behind creating a new term instead of using the well-established terminology that has been recognized for years in prominent scientific journals?

Q: Was the lowering of urine ammonia-nitrogen affected by increasing crystal amino acid levels in SBF diets?

Line 251: What were the causes for the improvement in digestibility in low-protein diets?

The improvement in digestibility may be due to the optimization of feed formulation and the use of alternative feed ingredients.

Q: How about the impact of increasing crystalline amino acids?

Line 325: This statement is incorrect, as the results did indicate an increase in backfat thickness and a reduction in lean muscle ratio (LMR) in pigs fed SBF. The study replaced not only soybean meal (SBM), but also a substantial amount of corn, which should not be overlooked. Furthermore, the feed-grade amino acids were also increased in the protein-decreasing diets. The effect of increasing feed-grade amino acids on nutrient digestibility, plasma amino acid levels, and urine ammonia nitrogen should be discussed.

Author Response

Line 19: Were amino acid data available?

A: The amino acid data can be found in Table 1.

Q: Was the concentration of each amino acid analysis result available?

A: The concentration of lysine, methionine, threonine and tryptophan was 78.8%, 99%, 99% and 98%, respectively.

Line 22: If it was not significant, was there no difference?

Not necessarily. Non-significant results do not always mean there are no differences. They may be due to factors such as sample size or variability. We hope to pay attention to these differences.

Q: The author must decide how to define the traits with a P value between 0.05 and 0.1. The tendency statement can be found in line 41.

A: We have rewritten this kind of tendency as: Compared with the CON group, lean meat percentage in the HSB and SBF groups showed a tendency to increase and decrease respectively (p = 0.09).

Line 42: What was used to replace SBM?

Soybean meal (SBM) was replaced by wheat, rice bran, wheat germ powder and crystal amino acid.

Q: Can this information be found in the abstract? The details regarding the ingredients used to replace SBM are crucial. In fact, these ingredients replace not only SBM but also significantly reduce the amount of corn. Therefore, the title of this manuscript should be more precise to reflect these key aspects.

A: Soybean meal (SBM) was replaced by ground corn, wheat, wheat bran, rice bran, and wheat germ powder, this was calculated by brill formulation software from USA.

Line 90: What was the health status of the pigs?

The health status of the pigs was good, as indicated by the lack of significant differences in serum biochemical parameters among the groups (Table 4).

Q: Serum biochemical parameters can be used as diagnostic tools, but other tools are also needed when assessing the health of animals. Was the medication record being reviewed?

A: In the experiment, the overall health status of the animals was mainly assessed through serum biochemical parameters.

Line 94: Did the pigs have free access to feed? What if pigs didn’t finish all the feed provided at each feeding time?

A: Pigs are allowed to feed freely. The feed allocation per pen is adjusted daily based on body weight, and the weight of the remaining feed is recorded before each new feeding.

Q: If pigs had free access to feed, why is the purpose of daily adjusted feed allocation?

A: Adjusting feed allocation simply refers to preventing situations such as insufficient feed reserves or excessive accumulation in the feed trough for a long time leading to mold.

Are the chemical analysis results of the diets available?

Yes, the chemical analysis results of the feed are provided in Table 1.

Q: Were dry matter, ether extract (EE), calcium (Ca), and phosphorus (P) analyzed in the results? It is difficult to believe that the replacement of SBM with these ingredients, while maintaining the same levels of inorganic sources of Ca and P, would result in exactly the same analyzed values across the three diets. Additionally, it seems unlikely that the crude protein (CP) results would be reported as integers.

A: The total calcium and total phosphorus in the formula were derived from different combinations of raw materials, and the digestibility of calcium and phosphorus in each raw material may be different, which may lead to this result.

Line 129: What concentration of H2SO4 was used? Which marker was used?

10% H2SO4 was added to facilitate nitrogen fixation. Ammonium sulfate was used as marker.

Q: The material and methods still lack information on digestive markers, and the analysis procedure for digestive markers is also required.

A: We Have added descriptions in Line 110, 140.

Line 134: How were Ca and P analyzed?

Calcium (Ca) and phosphorus (P) were analyzed using the methods described in the Materials and Methods section (AOAC methods).

Q: it still can’t be found in the Materials and Methods section.

A: We Have added descriptions in Line 144.

Line 140: Was it serum or plasma?

Serum and plasma samples were collected. The serum samples were used for the analysis of various biochemical parameters, while the plasma samples were used to determine the concentration of plasma free amino acids.

Q: Line 149: The author suggested blood was collected into plasma tubes.

 A: Thank you for pointing out this inappropriate description. Now we have added the following relevant description: To collect serum, draw blood into a plain tube, allow it to clot, and then centrifuge to separate the serum. For plasma collection, draw blood into an anticoagulant tube, centrifuge, and then separate the plasma. Store both serum and plasma samples at -20°C or -80°C.

Line 164: What was the experimental unit? What was the experimental design?

A total of 81 barrows and 81 gilts of 150 days old with similar body weights (103.64 ± 3.65 kg) were selected and randomly allocated into three groups with 3 replicates in each group and 18 finishing pigs in each pen. Three dietary formulations were established in this experiment (Table 1): (1)14% crude protein, and 14% soybean meal for the control group diet (CON); (2)13% crude protein, and 7% soybean meal for half soybean meal group diet (HSB); (3)12% crude protein, and 0% soybean meal for soybean meal free group diet (SBF).

Q: What was the experimental unit used for ANOVA? What was the experimental design of the statistical model?

A: The experimental unit used for ANOVA was the data of each duplicate with 18 finishing pigs, the experimental design of the statistical model was completely randomized design.

Line 173: The results of the feed conversion rate are concerning. If we divide ADFI by ADG, we get 3.597, 3.53, and 3.59 for CON, HSB, and SBF, respectively. I understand the shift in means after statistical analysis, but the relative differences between treatments should remain the same.

A: Due to the deviation of the average value when retaining decimal places, we revised the data of 3.60, 3.53, and 3.59 for CON, HSB, and SBF, respectively in the manuscript.

Line 191-193: The author indicated that pigs fed SBF had lower corn inclusion while maintaining comparable growth performance. This raises confusion regarding the original hypothesis, as it is unclear how the reduction in corn relates to the expected outcomes.

A: Although the amount of corn is reduced in formula, but the amount of wheat, rice bran and wheat germ meal increased respectively, and finally swine net energy in SBF was almost the same compared with other two groups. So, the expected outcomes were reasonable.

Line 217: If tendencies are meant to be reported, superscripts based on multiple comparisons to differentiate treatments are required.

A: Thank you very much for your suggestion, multiple comparisons are not reflected in the table, because tendencies do not have a significant impact on this experiment.

Q: If a tendency was observed but pairwise comparisons did not show a significant difference between treatments, the discussion of this tendency should be avoided. However, if pairwise comparisons did reveal a significant difference between treatments, it would be appropriate to discuss the tendency, provided that a superscript is included to indicate the statistical significance.

A: Differences in the text are indicated by superscript abc, while the absence of such notation means no significant difference.

By the way, what is slaughter rate, or is it referred to as yield percentage?

The slaughter rate is the percentage of yield.

Q: If this is meant to represent yield percentage, then what is the rationale behind creating a new term instead of using the well-established terminology that has been recognized for years in prominent scientific journals?

 A: Thank you for your suggestion, we have corrected it to "yield percentage."

Q: Was the lowering of urine ammonia-nitrogen affected by increasing crystal amino acid levels in SBF diets?

A: Yes, the absorption rate of crystal amino acid was higher that may lower urine ammonia-nitrogen in SBF diets.

Line 251: What were the causes for the improvement in digestibility in low-protein diets?

The improvement in digestibility may be due to the optimization of feed formulation and the use of alternative feed ingredients.

Q: How about the impact of increasing crystalline amino acids?

A: It could ensure that the SID amino acid content of each group was consistent after the soybean meal was replaced.

Line 325: This statement is incorrect, as the results did indicate an increase in backfat thickness and a reduction in lean muscle ratio (LMR) in pigs fed SBF. The study replaced not only soybean meal (SBM), but also a substantial amount of corn, which should not be overlooked. Furthermore, the feed-grade amino acids were also increased in the protein-decreasing diets. The effect of increasing feed-grade amino acids on nutrient dig.

A: Although the amount of corn is reduced in formula, but the amount of wheat, rice bran and wheat germ meal increased respectively, and finally swine net energy as well as SID amino acids in SBF was almost the same compared with other two groups. But the protein level in the SBF group was lower than CON and HSB groups. Maybe, this could lead to such an outcome.

Reviewer 2 Report

Comments and Suggestions for Authors

Thanks for the authors' responses. But the repetition rate of the article is more higher than the initial manuscript, which needs to be taken seriously.

Author Response

Thank you for your recognition and valuable suggestions. We will continue to make revisions to reduce the repetition rate.

Reviewer 3 Report

Comments and Suggestions for Authors

All comments were addressed.

Author Response

Thank you for your recognition and valuable suggestions.

Reviewer 4 Report

Comments and Suggestions for Authors

Ok with the global changes in the paper.

Author Response

(The authors gave the same response as above.)
